# Single molecule FRET reveals pore size and opening mechanism of a mechano-sensitive ion channel

Yong Wang[1,2], Yanxin Liu[1,2], Hannah A DeBerg[1,2], Takeshi Nomura[3], Melinda Tonks Hoffman[1,2], Paul R Rohde[3], Klaus Schulten[1,2], Boris Martinac[3,4], Paul R Selvin[1,2,5]*

[1]Department of Physics, University of Illinois at Urbana-Champaign, Urbana, United States; [2]Center for the Physics of Living Cells, University of Illinois at Urbana-Champaign, Urbana, United States; [3]Molecular Cardiology and Biophysics Division, Victor Chang Cardiac Research Institute, Sydney, Australia; [4]St Vincent's Clinical School, Faculty of Medicine, University of New South Wales, Sydney, Australia; [5]Center for Biophysics and Computational Biology, University of Illinois at Urbana-Champaign, Urbana, United States

**Abstract** The mechanosensitive channel of large conductance, which serves as a model system for mechanosensitive channels, has previously been crystallized in the closed form, but not in the open form. Ensemble measurements and electrophysiological sieving experiments show that the open-diameter of the channel pore is >25 Å, but the exact size and whether the conformational change follows a helix-tilt or barrel-stave model are unclear. Here we report measurements of the distance changes on liposome-reconstituted MscL transmembrane α-helices, using a 'virtual sorting' single-molecule fluorescence energy transfer. We observed directly that the channel opens via the helix-tilt model and the open pore reaches 2.8 nm in diameter. In addition, based on the measurements, we developed a molecular dynamics model of the channel structure in the open state which confirms our direct observations.

*For correspondence: selvin@illinois.edu

**Competing interests:** The authors declare that no competing interests exist.

## Introduction

Mechanosensitive (MS) channels are essential in both eukaryotes and prokaryotes (*Perozo, 2006*; *Árnadóttir and Chalfie, 2010*; *Haswell et al., 2011*). In eukaryotes, they are involved in diverse processes such as embryonic development, touch, pain, hearing, lung growth, and muscle homeostasis (*Hamill and Martinac, 2001*; *Chalfie, 2009*; *Árnadóttir and Chalfie, 2010*). In bacteria, they are 'safety valves', opening their pores to release the pressure to protect cells from hypo-osmotic shock (*Booth and Blount, 2012*). The rise in antibiotic resistance, and the crucial role MS channels play in bacterial adaptation, makes it important to understand the MS channels as potentially new drug targets (*Booth and Blount, 2012*).

When high pressure (~10 mN/m) causes the bacterial mechanosensitive channel of large conductance (MscL) to open, it forms a large, nonselective pore with a very high conductance (~3 nS) that is permeable to various ions and small organic osmolytes. In 1998, MscL from *Mycobacterium tuberculosis* in the closed state was crystallized by Rees and co-workers (*Chang et al., 1998*). They showed that MscL is a pentamer made up of five identical subunits (*Figure 1A,B*). Each subunit consists of one cytoplasmic α-helix (the CP domain) and two transmembrane α-helices (the TM1 and TM2 helices), which extend through the cell membrane and are joined by a periplasmic loop (*Figure 1B*). TM1 and TM2 are primarily responsible for gating; it has been shown that complete deletion of the CP domain does not change the gating parameters substantially (*Anishkin et al., 2003*).

**eLife digest** Bacterial cells are full of fluid, and they will burst if they are not able to respond to a build up of pressure. Fortunately, the membrane of a bacterial cell contains channels that can detect the increased mechanical stress on the cell membrane and then open to relieve the pressure.

In many bacterial cells, the last defence against the cell exploding is called the mechanosensitive channel of large conductance (MscL). This is made of five proteins, each of which consists of TM1 and TM2 helixes, which are responsible for opening and closing the channel. Two models have been proposed to explain how the channels are opened. In the barrel-stave model, the TM1 helix moves, while the TM2 helix remains stationary. This results in an open pore that is lined with TM1 and TM2 helixes in the same way that wooden staves line a barrel. In the helix-tilt model, both helixes tilt towards the membrane to open the channel.

Wang et al. have now used a technique called single-molecule fluorescence resonance energy transfer (FRET) to explore the structure of the open channel in *E. coli* in order to determine which model is correct. In this technique an individual channel is labeled with two different fluorescent molecules. By illuminating the channel with light of a wavelength that excites the first fluorescent molecule, and measuring the strength of the fluorescence from the second molecule, it is possible to work out the distance between the two molecules. From this, the structure of the channel and how it opens and closes can be explored.

Previous attempts to measure the diameters of open channels using fluorescence techniques have suffered from issues caused by the use of large numbers of fluorescent molecules. This has made it necessary to use computational modeling to extract the required data. By looking at a series of individual proteins, Wang et al. overcame these problems and found that the diameter of the fully open pore is 2.8 nm. The result provides strong support for the helix-tilt model.

Despite this progress, the open form of MscL has not been crystallized. This leaves two questions unanswered: what is the exact size of the open pore of MscL, and how does the channel open? Several techniques, for example, permeation of organic ions (*Cruickshank et al., 1997*), Electron paramagnetic resonance (EPR) (*Perozo et al., 2002a*, *2002b*) and ensemble fluorescence resonance energy transfer (FRET) (*Corry et al., 2005b*, *2010*) have attempted to measure the pore size. However, systematic errors likely result in an overestimation of (*Cruickshank et al., 1997*), an underestimation of (*Corry et al., 2005b*, *2010*), or an insensitivity to the requisite distances (*Perozo et al., 2002a*). For example, EPR was only able to establish that the open pore is >25 Å (11). Ensemble FRET, which yielded some insightful results, is potentially sensitive to larger distances (~80–100 Å) (*Roy et al., 2008*). However, due to multiple labeling, problems with protein clustering, and the need for Monte-Carlo simulations to extract distance information, there was much variability and uncertainty in the results (*Corry et al., 2005a*, *2005b*, *2010*).

Another important question is how the MscL channels open, that is how the helices rearrange upon channel activation (i.e., from the closed state to the open state). Currently, there exist two predominant models for the opening of MscL: the barrel-stave model and the helix-tilt model (*Figure 2*; *Perozo, 2006*). The barrel-stave model (*Figure 2C,D*) involves motion of the transmembrane helix 1 (TM1) with the transmembrane helix 2 (TM2) remaining stationary; the open pore is lined by both TM1 and TM2 and the helices are fairly vertical (where the membrane is horizontal). This model derives primarily from the number of transmembrane helices and the large size of the open pore. In contrast, the helix-tilt model (*Figure 2E,F*), which has been proposed more recently (*Sukharev et al., 2001a*, *2001b*; *Betanzos et al., 2002*), involves motion of TM1 and TM2, with both swinging away from the pore upon channel opening and both helices tilting toward the plane of membrane. Recent evidence from cysteine-crosslinking experiments, EPR experiments, and ensemble FRET experiments, argue in favor of the helix-tilt model (*Betanzos et al., 2002*; *Perozo et al., 2002a*; *Corry et al., 2010*).

In the present work, we focused on the transmembrane helices involved in the opening of MscL from *Escherichia coli* (EcoMscL), using a single-molecule fluorescence resonance energy transfer (smFRET). MscL channels were reconstituted in liposomes during smFRET measurements and thus the channels were in their quasi-native environment. In addition, although MscL is a pentamer, we utilized photobleaching to virtually sort out the population of molecules with a single donor and a single acceptor, allowing us to make accurate smFRET measurements. It is the first time that smFRET has been applied to

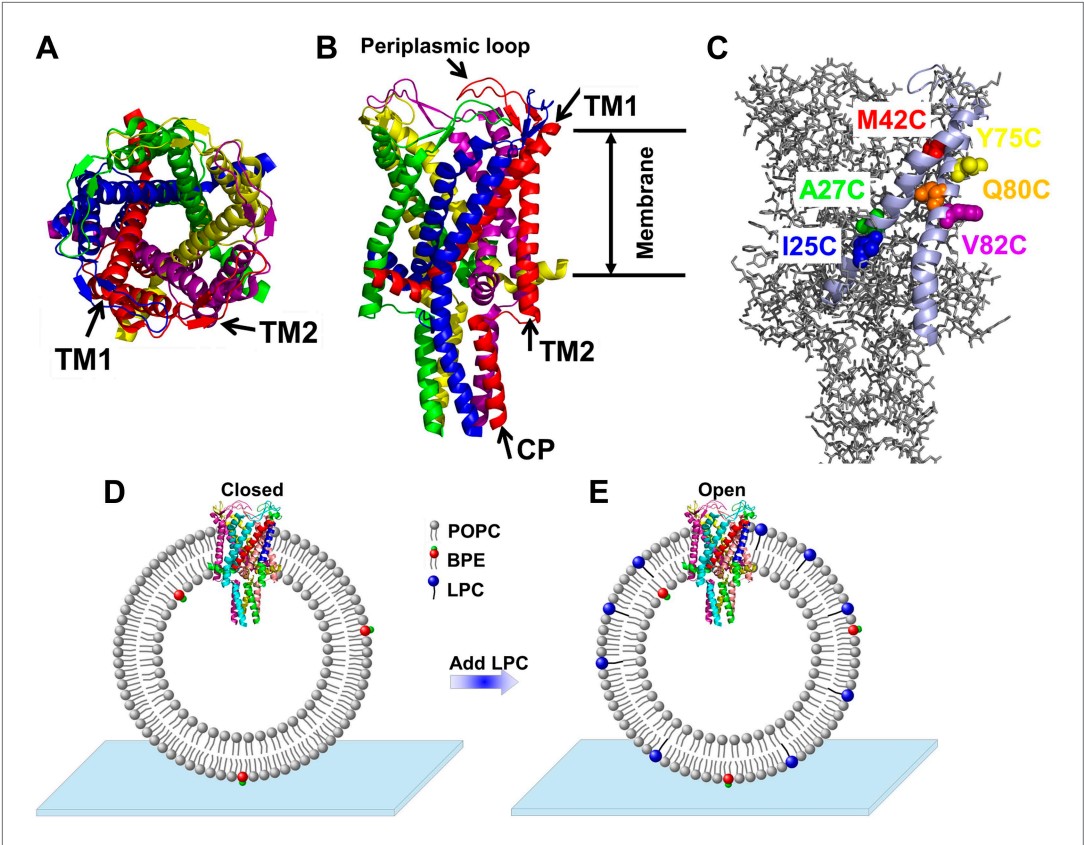

**Figure 1**. Cartoon representation of the structure of MscL in the closed conformation in the (**A**) top view and (**B**) side view (PDB ID: 2OAR [***Chang et al., 1998***; ***Steinbacher et al., 2007***]), and scheme of single molecule FRET setup. MscL is a homo-pentamer consisting of five identical subunits. Each subunit consists of one cytoplasmic α-helix (CP) and two transmembrane α-helices (TM1 and TM2), which extend through the cell membrane and are joined by a periplasmic loop (***Chang et al., 1998***). (**C**) Residues measured using smFRET. Three residues on each of the transmembrane helices (M42C, A27C and I25C on TM1; Y75C, Q80C and V82C on TM2) were chosen. Note that no residues on the CP were chosen because the complete deletion of the CP does not change the gating parameters substantially (***Anishkin et al., 2003***). (**D**) Labeled MscL proteins were reconstituted into liposomes, which were then immobilized on a coverslip and used for smFRET experiments. (**E**) The addition of LPC traps the protein in the open conformation (***Perozo et al., 2002b***).

liposome-reconstituted membrane proteins with more than three monomers. We measured movement of three residues on TM1 (M42C, A27C, and I25C; *Figure 1C*) and three residues on TM2 (Y75C, Q80C and V82C; *Figure 1C*), from which we determined not only the translational movements but also the tilting of each helix. We observed the tilting of the helices in a model-free fashion, arguing strongly in favor of the helix-tilt model. In addition, from the movement of the residue (I25C) right at the gating region, we determined directly that the open pore reaches 2.8 nm in diameter. Lastly, we developed a molecular dynamics model of the channel structure in the open state based on the smFRET results, while using the crystal structure of the protein in the closed state as a reference. The model of the open structure satisfies all the distance constraints measured from smFRET experiments. The developed open structure confirmed that the pore size of the fully open channel is 2.8 nm in diameter, achieved via the helix-tilt opening model.

## Results

### FRET efficiencies

Purified MscL mutants (*Figure 3—figure supplement 1*) were labeled with Alexa Fluor 488 (AF488) and Alexa Fluor 568 (AF568) and reconstituted into ~50 nm liposomes made of 1-palmitoyl-2-oleoyl-sn-glycero-3-phosphocholine (POPC) with 2% 1,2-dioleoyl-sn-glycero-3-phosphoethanolamine-N-biotinyl

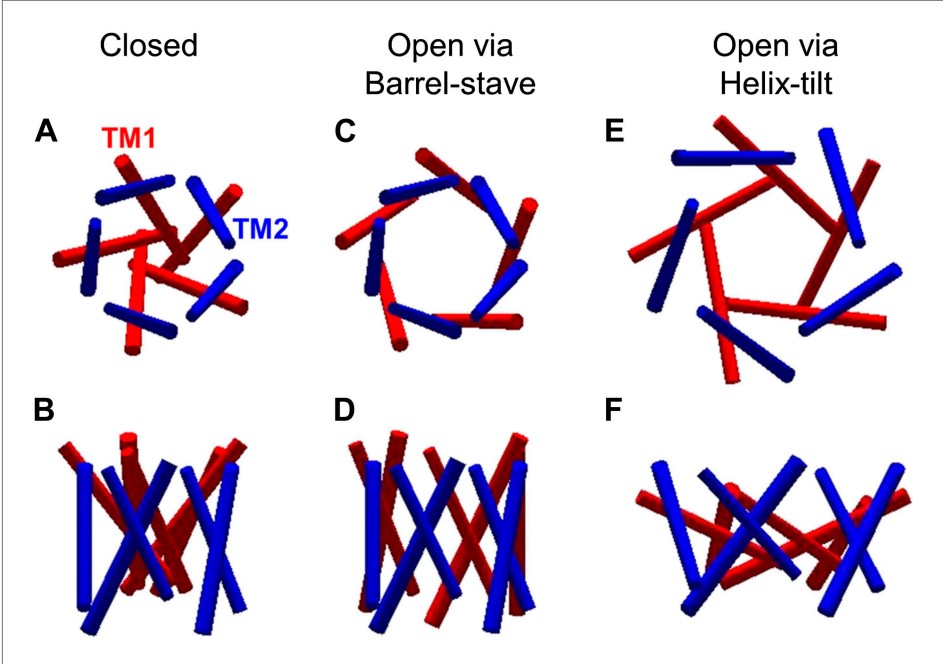

**Figure 2**. The opening models for MscL. The MscL opens from (**A** and **B**) the closed state, to (**C** and **D**) the open state via the barrel-stave model or (**E** and **F**) the open state via the helix-tilt model. The top figures (**A**, **C**, **E**) are top views and the bottom figures (**B**, **D**, **F**) are the side views. TM1 helices are shown in red while TM2 in blue. In the barrel-stave model (**C** and **D**), TM1 swings away from the pore center but TM2 remains stationary upon channel activation, generating an open pore lined by both TM1 and TM2 and the helices are more parallel to the membrane normal than the membrane plane. In the helix-tilt model (**E** and **F**), both TM1 and TM2 swing away from the symmetry axis and both helices tilt toward the plane of membrane.

(BPE) (*Figure 1D*). The liposomes were then immobilized on a glass coverslip, via biotin-avidin interaction, for smFRET measurements (*Figure 1D*). To access the open state of the channels, 1-oleoyl-2-hydroxy-sn-glycero-3-phosphocholine or lysophosphatidylcholine (LPC) of 25% molar ratio was added (*Perozo et al., 2002b*, *2002a*; *Corry et al., 2005b*, *2010*; *Figure 1E*) and incubated for >10 min before immobilization. Just before performing smFRET experiments, the fluorescence spectra of the samples (±LPC) were recorded with excitation at 488 nm to confirm that the channels open up with LPC by observing the shift in the FRET peaks. (The channel activity is also determined by observing the opening of the channels upon application of negative pressure [suction] to the patch pipette. The labeled proteins for patch-clamp experiments are from a different aliquot, although the same batch, of the labeled proteins for smFRET experiments). We emphasize that, although smFRET has been applied to study the conformational changes of channels and transporters (*Choi et al., 2010*; *Zhao et al., 2010*, *2011*; *Akyuz et al., 2013*), to our knowledge, it is the first time that smFRET has been used with channels reconstituted to liposomes.

Via smFRET measurements, we observed fluorescence intensity traces with one or two photobleaching steps (*Figure 3—figure supplement 2A,B*). This is the expected result because MscL is a homo-pentamer and the labeling of fluorophores is stochastic. The number of photobleaching steps tells the number of fluorophores attached to a channel. Only the traces showing a single photobleaching step in both the donor and acceptor channels, ensuring that only a single donor and/or acceptor fluorophore, were included in the analysis (*Figure 3—figure supplement 2A*). Donors were, in most cases, photobleached first, resulting in simultaneous dropping of the fluorescence intensities in both donor and acceptor channels (*Figure 3—figure supplement 2A,B*). Subtraction of the intensities before and after photobleaching gives the intensities of donor ($I_D$) and acceptor ($I_A$), which are used for the calculation of FRET efficiency. Note that the intensities, $I_D$ and $I_A$, automatically remove the direct excitation of acceptor (i.e., the leakage of acceptor emission in the donor channel). However, the leakage of donor emission in the acceptor channel is still present.

To measure this leakage, MscL channels were labeled with donors-only and the leakage coefficient (l) was measured experimentally: $l = I_D^A/I_D^D \approx 0.09$, where $I_D^A$ is the intensity of donor emission in the acceptor channel and $I_D^D$ is the intensity of donor emission in the donor channel. Furthermore, to determine the actual FRET efficiency, another instrumental correction was made through the correction factor γ, which accounts for the differences in quantum yield and detection efficiency between the donor and the acceptor. It was calculated as the ratio of change in the acceptor intensity, $\Delta I_A$, to the change in the donor intensity, $\Delta I_D$, upon acceptor photobleaching: from the traces where the acceptor photobleached first (*Roy et al., 2008*), we estimated the value $\gamma = \Delta I_A/\Delta I_D \approx 0.89 \pm 0.06$ (*Figure 3—figure supplement 2C*).

We analyzed a few hundred traces (varying between 134 and 577 traces) with single photobleaching steps in the absence and presence of LPC for each mutant (*Figure 1C*, *Figure 3*). Here we focus on the mutant M42C for the sake of illustration. For the single photobleaching steps of M42C, 428 and 577 traces, in the absence and presence of LPC, respectively, were analyzed. The corrected FRET efficiencies were calculated and their distribution was then plotted and fitted with Gaussians via maximum likelihood estimates, shown in *Figure 3A,B*, while the number of Gaussians was determined according to the corrected Akaike information criterion (AICc) and the Bayesian information criterion (BIC) (*Table 1*; *Akaike, 1974*; *Schwarz, 1978*; *Sugiura, 1978*). In the absence of LPC, we observed three peaks at E = 0.1, 0.28 and 0.63, respectively (*Figure 3A*). In the presence of LPC, the third peak showing the highest FRET efficiency diminishes, leaving mainly two Gaussians (E = 0.1 and 0.23, *Figure 3B*). This transition (i.e., the highest peak decreases and the lowest peak increases) is more obvious when we plotted the difference between the normalized FRET distributions ($\sum P^X = 1$, where **X** = + for in the presence of LPC and **X** = − for in the absence of LPC) under the two conditions, as shown in *Figure 3C*: after adding 25% LPC, the peak at E ~0.6 went away but the fraction of the peak at E ~0.1 built up. Note that the highest peak at E ~0.6 does not completely disappear in the presence of 25% LPC, which is consistent with (*Perozo et al., 2002b*).

In the absence of LPC, the existence of three peaks, rather than two peaks, can be explained by considering the effect of tethering on the liposome. As the MscL channel is a homo-pentamer, we initially expected two distances between donor and acceptor in each state ($R_n$ and $R_f$ in *Figure 4A*) and thus two peaks for the distribution of FRET efficiency, assuming that all the channels are closed in the absence of LPC. However, this assumption is not necessarily true, especially in our situation where liposomes are immobilized and the proteins are responsive to membrane tension. It had been predicted by theories and observed in experiments that immobilization of liposomes (or vesicles) results in significant membrane tension and possibly rupture (*Zhdanov et al., 2006*; *Chung et al., 2009*; *Serro et al., 2012*). In our experiments, the membrane tension is expected to be high, ~30–40 $k_BT$, due to the strong interaction between BPE and the surface via biotin-neutravidin (*Miyamoto and Kollman, 1993*; *Rico and Moy, 2007*). With such strong interaction, giant unilamellar vesicles ruptured spontaneously, as has been observed experimentally (*Chung et al., 2009*). The consequence is that some of the MscL channels switch to the open conformation upon the immobilization of the liposomes. (However, the fraction of open channels might be different for different mutants even if the membrane tension is similar). Therefore, the FRET histogram for the no-LPC sample includes a mixture of closed and open MscL channels. To test this hypothesis, control experiments were performed by increasing the fraction of BPE in the liposomes, guided by a theoretical prediction (*Zhdanov et al., 2006*): if the hypothesis was true, the membrane tension in the liposomes due to immobilization would be higher, more channels would open in the absence of LPC, and therefore the difference between the FRET histograms with and without LPC would be smaller. We varied the fractions of BPE in the liposomes from 2% to 16% and, indeed, observed that the difference between the samples with and without LPC decreases (*Figure 3D*). We quantified the difference by the (unscaled) variance, $\Delta P^2 = \sum_{i=1}^{N} \left(P_i^+ - P_i^-\right)^2$, where $P_i^X$ is the probability distribution of FRET efficiency, $\sum_{i=1}^{N} P_i^X = 1$. We observed that the variance $\Delta P^2$ decreased by 98%, from 0.084 to 0.002, when the fraction of BPE in the liposomes increased from 2% to 16%, supporting the hypothesis that the sample without LPC is a mixture of closed and open channels.

A simple estimation based on the crystal structure of MscL in the closed state (*Chang et al., 1998*) and previously predicted/estimated open pore-size (*Cruickshank et al., 1997*; *Perozo et al., 2002a*; *Corry et al., 2010*) indicates that it is likely that, due to limited resolution of FRET, the middle peak (E = 0.28) is an overlap of two peaks corresponding to $R_n$ of the open state ($R_{no}$) and $R_f$ of the closed state ($R_{fc}$) (for $R_n$ and $R_f$, *Figure 4A,B*). The geometry of the protein (i.e., fivefold symmetry) gives

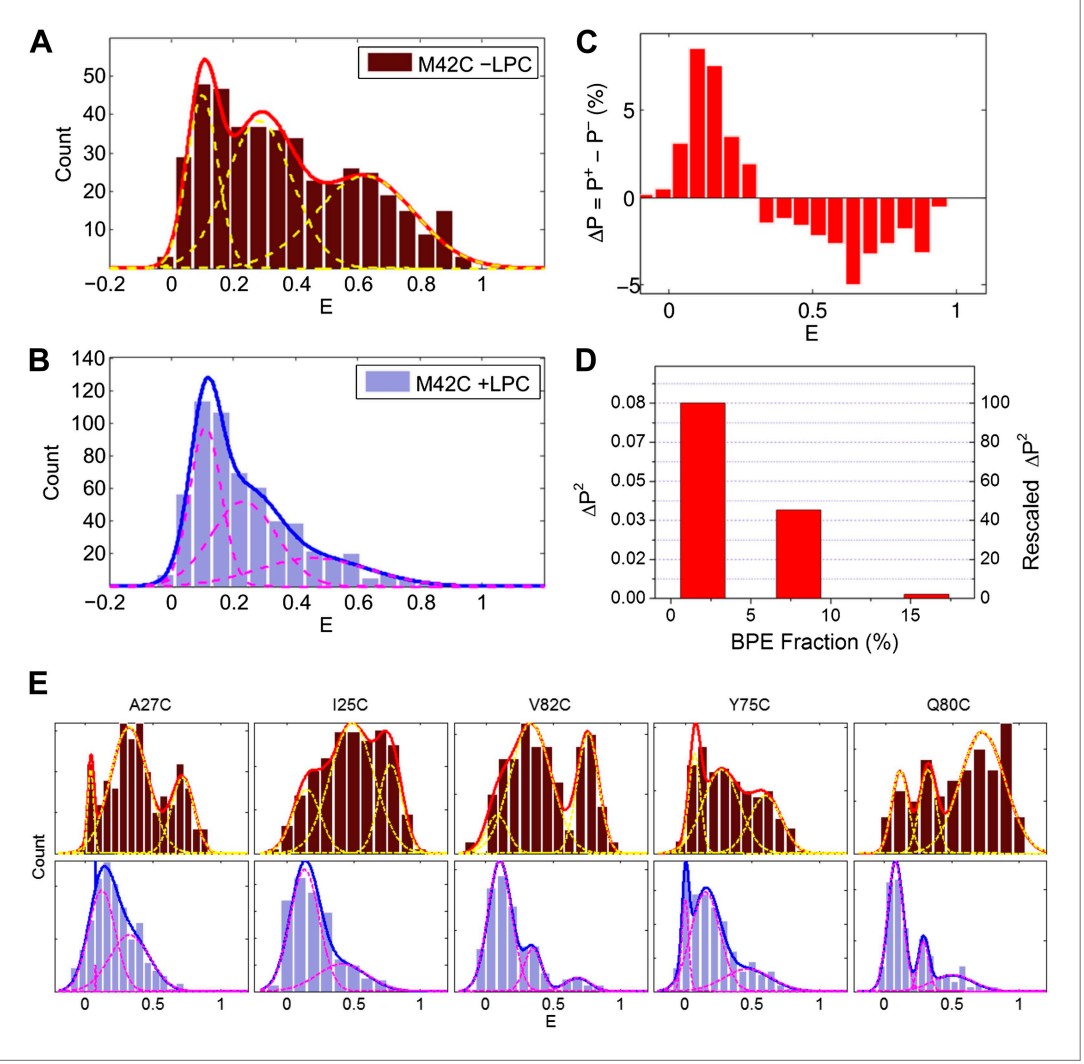

**Figure 3**. Single molecule FRET results for MscL M42C. The distribution of FRET efficiency of M42C in the (**A**) absence and (**B**) presence of LPC were plotted and fitted with Gaussians. (**C**) The difference between the normalized FRET distributions under the two conditions (±LPC), ΔP, emphasizes the diminishing of the third peak at E ~0.6 after adding LPC. (**D**) The variance between the normalized FRET distributions under the two conditions (±LPC), ΔP², decreases as the fraction of BPE in the liposomes is increased from 2% to 16%. (**E**) Histograms of FRET efficiencies in the absence (top row, −LPC) and presence (bottom row, +LPC) of LPC for the other five residues (I25C, A27C, Y75C, Q80C, and V82C) measured in the current study.

The following figure supplements are available for figure 3:

**Figure supplement 1**. Examples of FPLC traces and SDS-PAGE gel for purification of MscL.

**Figure supplement 2**. Examples of fluorescent intensity traces showing (A) a single photobleaching step, (B) multiple photobleaching steps, and (C) a single photobleaching step but the acceptor photobleached first.

**Figure supplement 3**. FRET efficiencies between non-neighboring subunits ($E_{f,o}$).

$R_n = D \cdot \sin\left(\dfrac{\pi}{5}\right)$ and $R_f = D \cdot \sin\left(\dfrac{2\pi}{5}\right)$ where D is the protein diameter. Take M42C as an example: the crystal structure gives the diameter of the protein in the closed state (using the $C_\alpha$ atoms), $D_c$ = 4.4 nm, resulting in $R_{fc} = D_c \cdot \sin\left(\dfrac{2\pi}{5}\right) = 4.2$ nm. The expected change in the protein size (ΔD) is from 2.5 to

**Table 1.** The best model of the fittings of FRET efficiency distribution were determined by calculating the corrected Akaike information criterion and Bayesian information criterion: $AIC_c = -2\ln L_M + 2k + \dfrac{2k(k+1)}{N-k-1}$, $BIC = -2\ln L_M + k\ln N$ where $L_M$ is the maximum likelihood by the model, k the number of parameters of the model, N the number of datapoints used in the fit (**Akaike, 1974**; **Schwarz, 1978**; **Sugiura, 1978**)

| # of fitting peaks | $-\ln L_M$ | AICc | BIC |
|---|---|---|---|
| M42C; No LPC | | | |
| 1 | −1.9 | 0.3 | 8.4 |
| 2 | −43.4 | −76.6 | −56.4 |
| 3 | −56.4 | **−96.4** | **−64.3** |
| 4 | −58.1 | −93.5 | −49.5 |
| M42C; With LPC | | | |
| 1 | −190.9 | −377.9 | −369.2 |
| 2 | −279.5 | −549.0 | −527.3 |
| 3 | −290.7 | **−565.1** | **−530.5** |
| 4 | −292.7 | −563.0 | −515.5 |

The lowest AICc and BIC values give the best fitting model: three peaks (bold values).

4 nm (**Cruickshank et al., 1997**; **Perozo et al., 2002a**; **Corry et al., 2010**), with the most recent report of 2.8 nm from ensemble FRET measurements (**Corry et al., 2010**). Then the expected $D_o = D_c + \Delta D$ is 7.2 nm, which gives $R_{no} = D_o \cdot \sin\left(\dfrac{\pi}{5}\right) = 4.2$ nm, exactly the same as $R_{fc}$ (The expected $R_{no}$ is 4.1–4.9 nm when taking into account the expected range from the literature). In the simple calculation above, the positions of $C_\alpha$ atoms for the estimations were used. However, the side chains of the residues and the attached fluorescent probes will add an additional length on the order of 2 nm, resulting in that the chance for $R_{fc}$ and $R_{no}$ to overlap is even higher. Furthermore, it has been observed that ≤30% of MscL are hexamers, instead of pentamers, in detergents such as n-Dodecyl-β-D-maltopyranoside (DDM) used in the current study (**Gandhi et al., 2011**). This would tend to 'smear' the middle peaks of FRET in the absence of LPC. Therefore, to be consistent and accurate, we always use the highest FRET efficiency ($E_{nc}$ and $E_{no}$) for the calculation of distance changes ($\Delta R_n$). On the other hand, we did find that all mutants give $E_{fo}$ measurements compatible (i.e., within error) with the final model (except that M42C is slightly off), as shown in **Figure 3—figure supplement 3**.

FRET between neighboring MscL channels on the same liposome had been a problem in previous ensemble FRET experiments. To decrease the likelihood of its happening, and to effectively solve the problem, we applied two strategies. First, we used 5% labeled channels together with 95% unlabeled ones for reconstitution in liposomes. As a result, we had 16x lower molar ratio of labeled proteins (pentamers) to lipids than that in the ensemble FRET experiments: 1:4000 vs 1:250 (**Corry et al., 2005b, 2010**), greatly reducing the likelihood of inter-molecular FRET. We found that adding a mixture of labeled and unlabeled protein helps to obtain no more than one fluorescent channel per liposome while ensuring efficient incorporation of channels into liposomes. In addition, only traces showing a single photobleaching step in both donor and acceptor channels were included in analysis, which helps further removing the FRET between neighboring MscL channels in the analysis. These strategies reduce significantly the likelihood that energy transfer happens between two adjacent channels even in the presence of MscL clustering, simplifying the interpretation of FRET results.

Another note is that we used maximum-likelihood estimation (MLE) (**In Jae, 2003**) for peak fitting. This method was chosen particularly because it does not require binning the data before fitting. Although there are mathematical ways for selection of 'good' bin sizes (**Shimazaki and Shinomoto, 2007**), the selection of bin size is, in practice, subjective, and the peaks derived can be affected with different bin sizes. After MLE fitting, we then bin the data and plot the histograms for the sake of presentation purpose. How the data is binned does not change the fitting parameters.

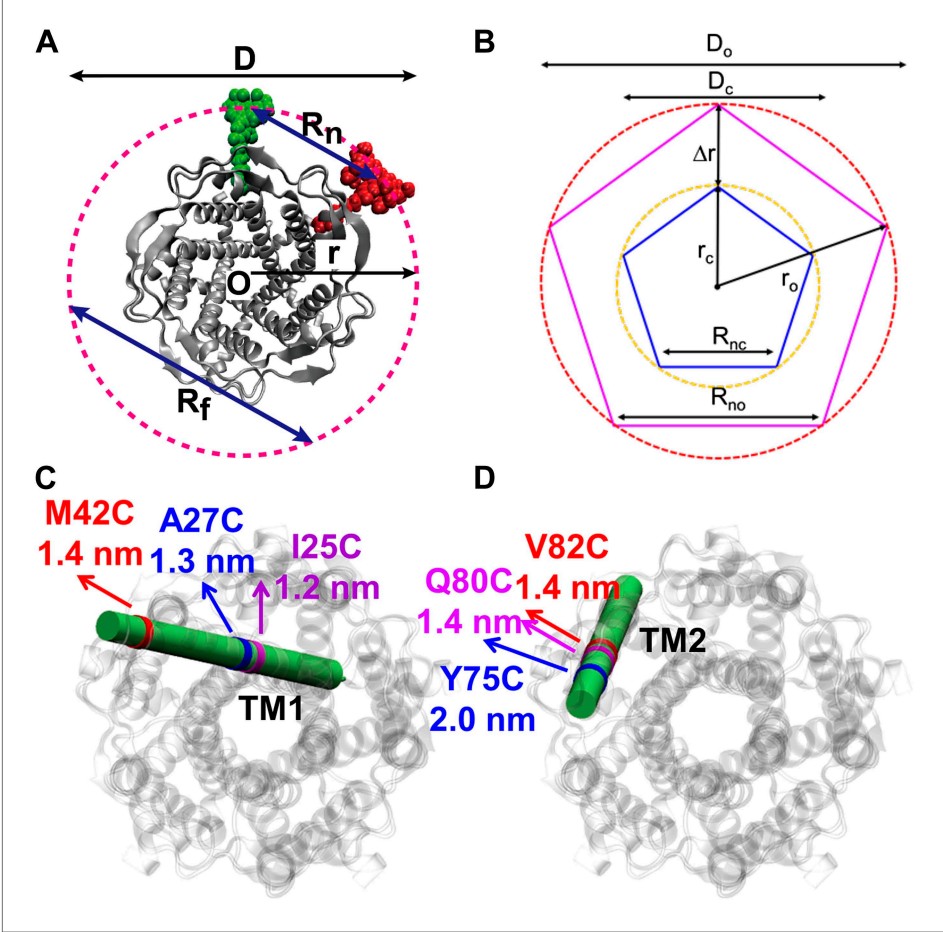

**Figure 4**. Movement of residues. (**A**) Each residue (highlighted in green) defines a circumcircle (dashed red circle) of radius r (or diameter D, where D, as shown, is $D_{closed}$, although upon opening would be $D_{open}$), centered at the pore center (*O*). Upon channel activation, the protein expands (radius changes from $r_{close}$ to $r_{open}$), or equivalently, the residue moves by $\Delta r = r_{open} - r_{close}$, measured from the pore center (*O*). (**B**) Sketch of MscL from closed state (blue pentagon) to open state (purple pentagon). The residue of interest (vertices of the pentagons) moves $\Delta r$ from the pore center. (**C** and **D**) Translational movements ($\Delta r$) of residues on TM1 and TM2 measured via smFRET. All the residues move away from the pore center, arguing in favor of the helix-tilt model.

The following figure supplements are available for figure 4:

**Figure supplement 1**. Molecular structures of fluorophores used in the experiments.

**Figure supplement 2**. Geometric analysis of the distances of interest while taking into account the finite size of fluorescent probes and the breaking of fivefold symmetry of the protein due to attachment of probes.

**Figure supplement 3**. Effect of hexameric MscL in protein preparation.

## Measurement of Förster radius, $R_0$

The Förster radius ($R_0$) for AF488 and AF568 is calculated by means of $R_0 \propto (\kappa^2 Q_D)^{1/6}$ (*Förster, 1948*; *Iqbal et al., 2008*). Because $\kappa^2$ and $Q_D$, can be environmentally sensitive, we measured the quantum yield and orientation factor for the fluorophores conjugated to each and every channel mutant (*Fery-Forgues and Lavabre, 1999*; *Lakowicz, 1999*; *Figure 5*). The quantum yields of AF488 conjugated to various MscL mutants are summarized in *Table 2*; *Figure 5A*, corrected for polarization effects (*Fery-Forgues and Lavabre, 1999*; *Lakowicz, 1999*). It is noted that the fluorophores used in the current study are mixtures of 5' and 6' isomers. However, it was expected that this will not affect the results because (1) they have successfully been used in many smFRET studies (*Marras et al., 2002*; *Yin et al.,*

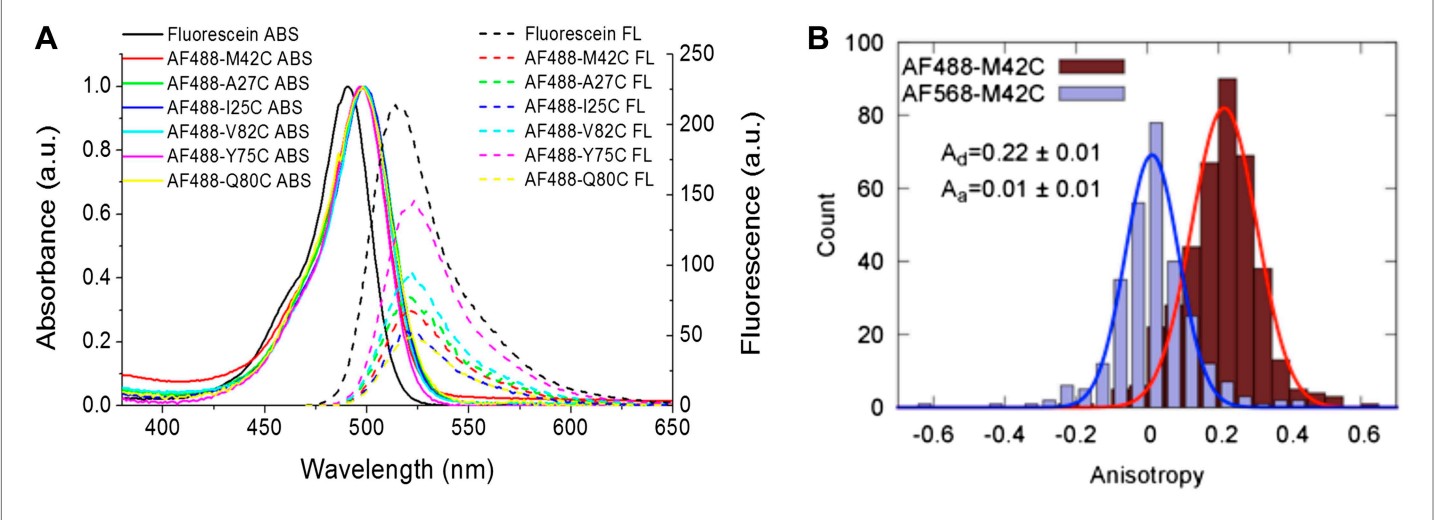

**Figure 5**. Measurement of $R_0$. (**A**) Absorbance and fluorescence spectra of AF488-MscL and fluorescein (as a standard), used to determine the quantum yield of AF488 conjugated to MscL mutants. (**B**) Anisotropy of AF488 and AF568 conjugated to MscL mutant (M42C), corrected for the intrinsic polarization properties of the microscope, and for the high numerical aperture of the objective.

The following figure supplements are available for figure 5:

**Figure supplement 1**. Anisotropy of AF488 and AF568 conjugated to MscL mutants, corrected for the intrinsic polarization properties of the microscope, and for the high numerical aperture of the objective.

*2005*; *Jäger et al., 2006*; *Granier et al., 2007*; *Majumdar et al., 2007*); (2) the chromophores of the isomers are exactly the same while the only difference between the isomers lies in where the linker of carbon-chain [$(CH_2)_5NHCO$] is attached; (3) we examined the molecular structures of the probe-isomers and confirmed that the difference in molecular size is <5% between isomers (*Figure 4—figure supplement 1*).

The orientation factor $\kappa^2$, was determined by measuring the anisotropy of the conjugated fluorophores (*Table 2*; *Figure 5B*, *Figure 5—figure supplement 1*). The anisotropy of both donor and acceptor for most residues is <0.2 and therefore $\kappa^2$, in fact, is close to 2/3 (*Clegg, 1992*; *Andrews and Demidov, 1999*; *Roy et al., 2008*). Nonetheless, we calculated the maximum possible errors in $R_0$ due to anisotropic orientation of the dyes (*Table 2*; *Figure 5B*, *Figure 5—figure supplement 1*); the actual errors in $R_0$ should be much smaller. Another source of error in $R_0$ lies in the measurements of $Q_D$, which were performed for AF488-MscL in detergent (PBS + 1 mM DDM), which was not exactly the same

**Table 2.** Measurements of smFRET experiments

| Residue | Helix | $Q_d$ | $A_d$ | $A_a$ | $R_0$ (nm) | $E_{nc}$ | $E_{no}$ | $\Delta R_n$ (nm) | $\Delta r$ (nm) | $\Delta D$ (nm) |
|---------|-------|-------|-------|-------|------------|----------|----------|-------------------|-----------------|-----------------|
| 42 | TM1 | 0.33 | 0.22 | 0.01 | $5.5^{+0.4}_{-0.3}$ | 0.63 | 0.23 | $1.7^{+0.7}_{-0.5}$ | $1.4^{+0.6}_{-0.4}$ | $2.8^{+1.1}_{-0.8}$ |
| 27 | TM1 | 0.28 | 0.12 | 0.09 | $5.3^{+0.4}_{-0.3}$ | 0.72 | 0.33 | $1.5^{+0.6}_{-0.4}$ | $1.3^{+0.5}_{-0.3}$ | $2.5^{+0.9}_{-0.6}$ |
| 25 | TM1 | 0.42 | 0.19 | 0.06 | $5.7^{+0.5}_{-0.4}$ | 0.78 | 0.42 | $1.4^{+0.6}_{-0.5}$ | $1.2^{+0.6}_{-0.4}$ | $2.4^{+1.1}_{-0.8}$ |
| 75 | TM2 | 0.62 | 0.19 | 0.11 | $5.6^{+0.6}_{-0.5}$ | 0.60 | 0.16 | $2.4^{+1.0}_{-0.7}$ | $2.0^{+0.8}_{-0.6}$ | $4.0^{+1.6}_{-1.2}$ |
| 80 | TM2 | 0.22 | 0.18 | 0.09 | $5.1^{+0.5}_{-0.3}$ | 0.72 | 0.29 | $1.6^{+0.7}_{-0.4}$ | $1.4^{+0.6}_{-0.4}$ | $2.7^{+1.1}_{-0.8}$ |
| 82 | TM2 | 0.39 | 0.24 | 0.13 | $6.1^{+0.7}_{-0.6}$ | 0.76 | 0.35 | $1.6^{+0.9}_{-0.8}$ | $1.4^{+0.8}_{-0.6}$ | $2.7^{+1.5}_{-1.3}$ |

$Q_d$ is the quantum yield of donor (AF488) after conjugation to each MscL mutant. $A_d$ and $A_a$ are the anisotropy of donor (AF488) and acceptor (AF568) after conjugation, respectively. $R_0$ is the Förster radius. $E_{nc}$ and $E_{no}$ are the FRET efficiencies in the closed and open states, respectively. $\Delta R$ is the change in the distances between donor and acceptor ($\Delta R_n = R_{no} - R_{nc}$, **Figure 4A,B**). $\Delta D$ is the change of the protein diameter ($\Delta D = D_o - D_c$). $\Delta r$ is the translational movement of the residue, measured from the pore center, $\Delta r = \Delta D/2$. Note that the errors shown in the table are the maximum possible errors due to anisotropic orientation of the dyes. The actual errors are expected to be much smaller.

environment for fluorophore-MscL conjugates in smFRET experiments (incorporated in liposomes in PBS), although the buffer was kept the same. Furthermore, it is possible that the addition of LPC and the conformational change of MscL changes $Q_D$ as well, resulting in additional errors in $R_0$ and in the distances calculated below.

It is noticed that the anisotropy of the acceptor is consistently lower than that of the donor although the acceptor is larger. This could be attributed to many factors, as stated by the Perrin equation (*Perrin, 1926*): $A_0/A = 1 + 6D\tau$, including the rotational diffusion coefficient (D), fluorescence lifetime ($\tau$) and, more significantly, the fundamental anisotropy ($A_0$) which varies according to wavelength (*Weber and Shinitzky, 1970*; *Lakowicz, 1999*). The anisotropies given in *Table 2* were measured at the wavelengths where smFRET experiments were performed.

## Strategy to avoid the effect of the finite size of fluorescent probes

We note that the finite size and length of the fluorescent probes brings additional difficulty to converting FRET measurements to the estimation of distances and therefore to structural modeling. To overcome this difficulty, our strategy is to focus on the movement of residues, instead of the absolute distances, and then to develop the structural model based on the *changes* of distances.

For illustration purpose, we set up the general scene in *Figure 4—figure supplement 2*. The pentameric MscL channel (light blue), centered at point O, is labeled with two probes (dark blue) on two cysteine residues (specified by residue #), while the other three residues (with the same residue # but on different subunits) remain empty/unlabeled (dashed green). (Here the breaking of symmetry is taken into account in this general case). To emphasize the size and length of the probes, filled green and red circles are used to indicate the actual chromophores. Then the actual chromophores of the labeled probes (green and red circles) and the center of the protein (O) define a pentagon (dotted orange pentagon, with a side length of R) and a circumcircle (black circle, with a diameter of D, or a radius of r = D/2). We call R, D and r apparent distance/diameter/radius because they could be measured/calculated from FRET experiments (with corrections) and they gave the apparent size of the protein (i.e., the protein appears to have a radius of r based on FRET experiments). There is another set of distances, which are more relevant to the protein and to the MD simulations for the structural model. We call this set of distances true values. For example, the true radius of the protein ($r_t$) (referring to a specific residue) is defined as the distance between the Ca atom of the residue and the center of protein (O). We note that the exact atom chosen for the definition of the true radius ($r_t$) does not matter.

Generally $r = r_t + r_p \neq r_t$ due to the finite size of the probes ($r_p \neq 0$). As a result, converting the FRET measurements into a structural model is not straightforward. However, if the size of the probes does not change (i.e., $\Delta r_p = 0$) upon channel activation, we then have $\Delta r = \Delta r_p + \Delta r_t = \Delta r_t$ (similarly $\Delta D = \Delta D_t$). We believe $\Delta r_p = 0$ is a reasonable assumption for the following reasons: (1) no chemical reactions are happening for the probes and thus the structures of the probes do not change before and after channel opening; (2) anisotropy measurements show that the orientation of probes are not constrained significantly. As a result, the change in the apparent distances is the same as the change in the true distances. In other words, the movement of residues (in the radical direction, $\Delta r_t$) can be obtained from the FRET measurements, even if the sizes and lengths of probes are nonzero.

A note to make is that we have assumed that donors and acceptors have similar sizes in the argument above. This assumption could be justified by the molecular structures of the probes (shown in *Figure 4—figure supplement 1*), which shows that the difference between the donor-size and acceptor-size is <2%. On the other hand, a caveat is that, although not likely, there are possible situations where the size of the probe can change (i.e., $\Delta r_p \neq 0$) upon channel opening, due to, for example, steric hindrance.

To conclude, the finite size of probes ($r_p$ ~1.7 nm) brings additional difficulties to converting FRET measurements to estimation of distances: FRET results gave the distances between the chromophores of donors and acceptors, which is different from the distances between the $C_\alpha$ atoms of residues of interest. However, on the other hand, the *movement* of the residues (or the movement of the $C_\alpha$ atoms of the residues) in the radial direction is the same as the *movement* of the chromophores *assuming* that the size of the probes does not change (i.e., $\Delta r_p = 0$) upon channel opening. We also note that, although the fivefold symmetry is broken due to the binding of one donor and one acceptor per pentamer, the geometric construction will not be affected. From here on, we use $\Delta r$ (and $\Delta D$) for the movement of residues (and the change of the protein diameter) without any subscript.

## Estimation of residue movements

We measured the change of FRET efficiency of MscL before and after channel activation using smFRET. For example, for M42C, the FRET efficiency changed from 0.63 (closed state) to 0.23 (open state). We also determined experimentally the Förster radii ($R_0 = 5.5^{+0.4}_{-0.3}$ nm for M42C). This permitted us to estimate the change in the distance between donors and acceptors from the closed to open states (*Figure 4*), $\Delta R_n = R_{no} - R_{nc} = R_0 (E_{no}^{-1} - 1)^{-1} - R_0 (E_{nc}^{-1} - 1)^{-1}$ ($\approx 1.7$ nm for M42C). We emphasize that some of the distances between fluorophores ($R_{no}$ and $R_{nc}$ in *Table 3*) are indeed out of the sensitive range of EPR measurements, making FRET a more suitable technique in this context.

As illustrated in the previous section, we focus on the more relevant distance of interest: the movement of the residue away from the pore center, $\Delta r$ (*Figure 4B*), or the change of protein diameter measured from the residue, $\Delta D$. Because of the fivefold symmetry of the MscL channel, $\Delta D$ and $\Delta r$ can be calculated readily according to $\Delta D = \Delta R_n / \sin(\pi/5) \approx 2.8$ nm, which yields $\Delta r = \Delta D/2 \approx 1.4$ nm (for M42C). The $\Delta r$ values of the residues are summarized in *Table 2*. This value is above 2.5 nm, a lower bound predicted by EPR experiments (*Perozo et al., 2002a*), but larger than $\Delta D$ obtained from the previous ensemble FRET measurement: $\Delta D_{M42C} = 2.8$ nm (smFRET) vs $\Delta D_{M42C} = 1.6$ nm (ensemble FRET) (*Corry et al., 2005b*, *2010*). We emphasize that the measurements of two more residues (I25C and A27C) in the current study were also reported previously (*Corry et al., 2010*). Our results are close to the values in their simulations ($\Delta D_{I25C} = 2.4$ vs 2.5 nm; $\Delta D_{A27C} = 2.5$ vs 2.6 nm) but differ significantly from the values measured directly from ensemble FRET experiments ($\Delta D_{I25C} = 2.4$ vs 0.2 nm). It should be noted that ensemble experiments gave inconsistent measurements for $\Delta D_{I25C} = 0.2$ nm and $\Delta D_{A27C} = 2.9$ nm, although the two residues are close. In contrast, smFRET results show that $\Delta D_{I25C} = 2.4$ nm is similar to $\Delta D_{A27C} = 2.5$ nm. This clearly demonstrates the advantage of smFRET.

We note that fluorophores/linkers at different residues are likely to be constrained differently. Furthermore, how they are constrained differently is not clear, partly due to the unavailability of the crystal structure of EcoMscL. However, certain residues are in agreement between the EcoMscL and the MtMscL (*Perozo et al., 2001*). Nevertheless, the distances between donors and acceptors are *not* good to compare for different residues of EcoMscL. A more reasonable way is to compare the *changes* of distances, that is, the movements of residues.

The calculations above were performed with the assumption that EcoMscL are pentamers. However, it is noted that a mixture of hexamers and pentamers were observed in certain detergents for EcoMscL (*Gandhi et al., 2011*) (≤30% hexamers in DDM). In our experiments, we ran FPLC (Superdex 200 10/300 GL column) for our MscL proteins and used the proteins from a single peak and checked with SDS-PAGE (*Figure 3—figure supplement 1*). In addition, in the smFRET experiments, we reconstituted the channels into liposomes, different from proteins in detergents where a mixture of hexameric channels and pentameric channels were observed (*Gandhi et al., 2011*). Third, as the conductance of the mutants (from electrophysiological recordings) used in this study agrees with that of MscL pentamers (*Figure 4—figure supplement 3C*), it is likely that we are reconstituting MscL pentamers into liposomes. Nonetheless, the possibility of having a (small) portion of hexamers in the sample in

**Table 3.** Measurements of smFRET experiments

| Residue | Helix | $E_{nc}$ | $E_{no}$ | $R_{nc}$ (nm) | $R_{no}$ (nm) | $\Delta R_n$ (nm) | $\Delta r$ (nm) | $\Delta D$ (nm) |
|---|---|---|---|---|---|---|---|---|
| 42 | TM1 | 0.63 | 0.23 | $5.0^{+0.4}_{-0.3}$ | $6.7^{+0.5}_{-0.4}$ | $1.7^{+0.7}_{-0.5}$ | $1.4^{+0.6}_{-0.4}$ | $2.8^{+1.1}_{-0.8}$ |
| 27 | TM1 | 0.72 | 0.33 | $4.6^{+0.3}_{-0.2}$ | $6.0^{+0.4}_{-0.3}$ | $1.5^{+0.6}_{-0.4}$ | $1.3^{+0.5}_{-0.3}$ | $2.5^{+0.9}_{-0.6}$ |
| 25 | TM1 | 0.78 | 0.42 | $4.6^{+0.4}_{-0.3}$ | $6.0^{+0.5}_{-0.4}$ | $1.4^{+0.6}_{-0.5}$ | $1.2^{+0.6}_{-0.4}$ | $2.4^{+1.1}_{-0.8}$ |
| 75 | TM2 | 0.60 | 0.16 | $5.7^{+0.5}_{-0.4}$ | $8.1^{+0.8}_{-0.6}$ | $2.4^{+1.0}_{-0.7}$ | $2.0^{+0.8}_{-0.6}$ | $4.0^{+1.6}_{-1.2}$ |
| 80 | TM2 | 0.72 | 0.29 | $4.4^{+0.4}_{-0.3}$ | $5.9^{+0.5}_{-0.4}$ | $1.6^{+0.7}_{-0.4}$ | $1.4^{+0.6}_{-0.4}$ | $2.7^{+1.1}_{-0.8}$ |
| 82 | TM2 | 0.76 | 0.35 | $4.7^{+0.5}_{-0.5}$ | $6.2^{+0.7}_{-0.6}$ | $1.6^{+0.9}_{-0.8}$ | $1.4^{+0.8}_{-0.6}$ | $2.7^{+1.5}_{-1.3}$ |

$E_{nc}$ and $E_{no}$ are the FRET efficiencies in the closed and open states, respectively. $R_{nc}$ and $R_{no}$ are the distances between donor and acceptor. $\Delta R$ is the change in the distances between donor and acceptor ($\Delta R_n = R_{no} - R_{nc}$). $\Delta D$ is the change of the protein diameter ($\Delta D = D_o - D_c$). $\Delta r$ is the translational movement of the residue, measured from the pore center, $\Delta r = \Delta D/2$. Note that the errors shown in the table are the maximum possible errors due to anisotropic orientation of the dyes. The actual errors are expected to be much smaller.

smFRET experiments could not be excluded completely. To estimate the introduced uncertainties, we investigated quantitatively the effect of the presence of hexamers via numerical simulations and found that the main effect of the presence of hexamers is not the shift of the peak center but the broadening of peak width (*Figure 4—figure supplement 3A-B*). If the sample was 100% hexamers, the diameters of the protein, as well as the pore size, will be greater by a factor of $\sin(\pi/5)/\sin(\pi/6)-1 \approx 17.6\%$. In the presence of 30% hexamers, as observed in (*Gandhi et al., 2011*), the calculations would be off by about 7.5%.

Because the size of both Alexa fluorophores is significant (~1.7 nm), it is possible that the attachment of the fluorophores to MscL channel results in various effects on the protein and on the FRET measurements. For example, the presence of the fluorophores might sterically hinder the conformational change of the proteins and prevent them from opening or closing. On the other hand, the steric hindrance might constrain the orientation of fluorophores, affect the relative orientation between the fluorophores and therefore add more errors on the distances converted from FRET efficiencies. In addition, the insertion of fluorophores to the protein might force the channel to be in a state different from the fully closed state, resulting in the distance change measurement is underestimated. However, we would like to emphasize that the expected effect is insignificant for the following reasons. First, if the insertion of fluorophore would result in significant steric hindrance on the protein, it is expected that the labeling is difficult (i.e., it takes much more effort for the fluorophores to be attached due to the steric hindrance). In other words, it is expected that steric hindrance is not significant on the mutants that are labeled well. More importantly, the channels after being labeled with AF488 and AF568 were confirmed to be functional by both ensemble FRET experiments (by observing the shift in the FRET peak) and patch-clamp measurements (by observing the opening of the channels upon application of negative pressure to the patch pipette) as shown in *Figure 6* and previous publications with the same fluorophores (*Corry et al., 2010*).

## Computational MscL opening model

With smFRET, we measured the movements of three residues on TM1 (M42C, A27C, and I25C) and three on TM2 (Y75C, Q80C and V82C) summarized in *Table 2* and *Figure 4C,D*. We observed *directly* and reliably for the first time, that both TM1 and TM2 swing away from the pore, supporting the helix-tilt model. Note that, among the three residues on each helix, two sites were very close to each other (A27C and I25C on TM1, Q80C and V82C on TM2). They were chosen purposefully to be close; they served as consistency checks and confirmed that our smFRET measurements are accurate (*Table 2*). In addition, the top of both helices (periplasmic side, *Figure 1B,C*; residues 42 on TM1 and 75 on TM2) moves further than the bottom (1.4 nm vs 1.2 nm for TM1 and 2.0 nm vs 1.4 nm for TM2), indicating that rotational tilting of the helices (toward the membrane plane) is involved. We emphasize that it is the first *direct* (model-free) observation of both TM1 and TM2 swinging away from the pore center and of the tilting of the transmembrane helices. Therefore it is the first *direct* observation in favor of the helix-tilt model.

To quantitatively investigate in detail how the MscL channel opens (i.e., how the helices move and rotate upon opening), we developed a computational model for the open structure of the MscL, starting from the crystal structure of MscL in the closed state (PDB: 2OAR) (*Chang et al., 1998*; *Steinbacher et al., 2007*) and employing the measured residue movements. For this purpose, we performed MD simulation with distance constraints (*Brünger et al., 1986*; *Trabuco et al., 2009*) (i.e., a virtual spring, *Figure 7—figure supplement 1*) using NAMD 2.9 (*Phillips et al., 2005*). Although similar modeling attempts have been made by *Corry et al. (2010)* and *Deplazes et al. (2012)* by using distance changes measured from ensemble FRET, we would like to emphasize that all smFRET measurements were used for the simulation while previously only a selected subset of ensemble data were used (as other data were not consistent with the resultant model) (*Corry et al., 2010*). For each measured residue, ten virtual springs were placed, five springs between the central carbon atom $C_\alpha$ of identical residues (highlighted green in *Figure 7—figure supplement 1*) from adjacent monomers (red springs in *Figure 7—figure supplement 1*) and five springs between the $C_\alpha$ of identical residues from non-adjacent monomers (yellow springs in *Figure 7—figure supplement 1*). The virtual springs were not applied to side chains because the flexibility of side chains likely introduces errors under large forces in the modeling process. The equilibrium lengths of the springs were chosen by adding the distance changes measured from smFRET to the equilibrium distances seen in the closed state, thereby, opening the crystal structure of *M. tuberculosis* MscL (PDB: 2OAR) (*Chang et al., 1998*; *Perozo et al., 2001*;

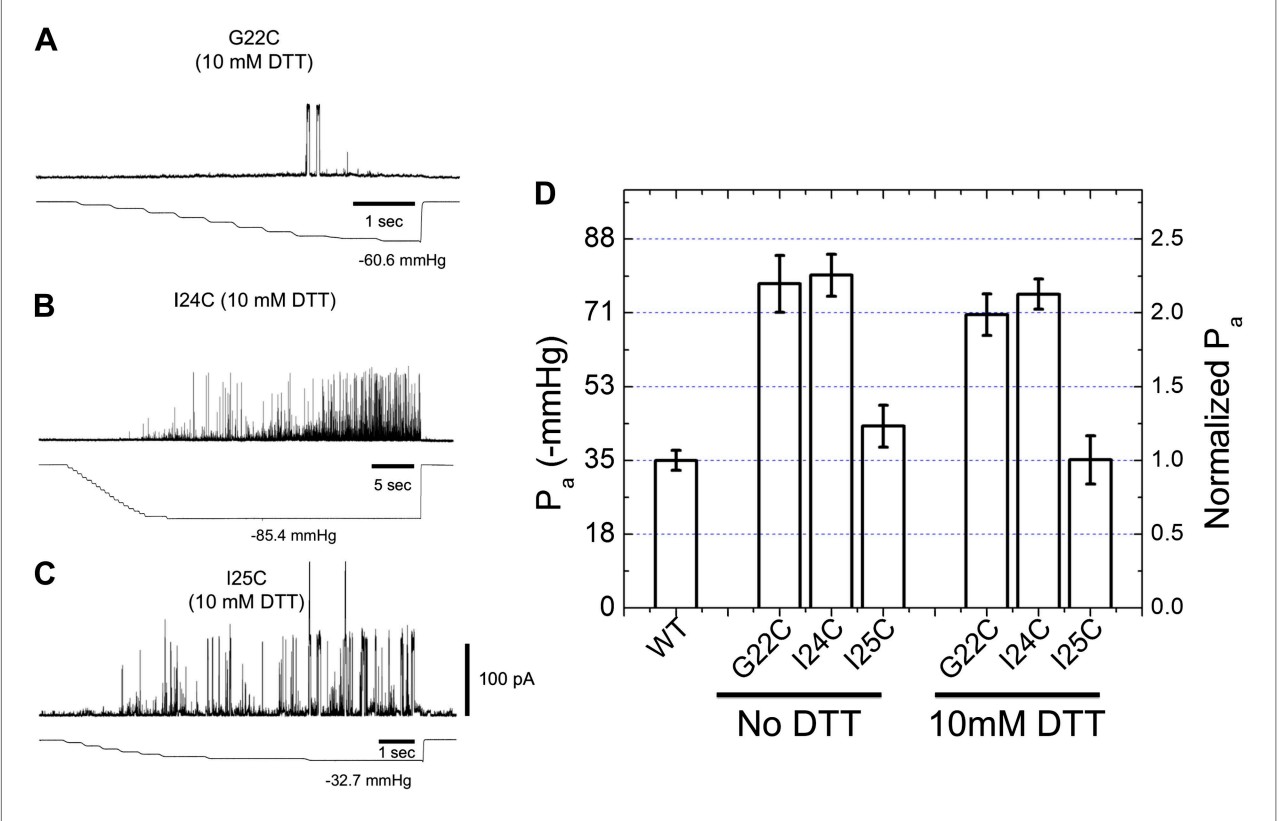

**Figure 6**. Activation thresholds, $P_a$, of MscL mutants at the proximity of the narrowest pore constriction. The activation thresholds were determined by electro-physiological recordings by patch-clamping without and with 10 mM DTT. Three recordings in the presence of DTT are shown as examples: (**A**) G22C, (**B**) I24C, and (**C**) I25C. (**D**) Comparison of the mutants with the wild type (WT) shows that the thresholds for mutants G22C and I24C are more than twice higher than the wild type, indicating the function of the channel was affected by the mutations. This was also observed via ensemble and single molecule FRET experiments. However, the mutation I25C does not affect the gating parameter substantially.

The following figure supplements are available for figure 6:

**Figure supplement 1**. Positions of MtMscL mutants at the proximity of the narrowest pore constriction.

*Steinbacher et al., 2007*). In the simulation, the virtual springs pushed corresponding residues from the distance in the closed state to the equilibrium length in the open state. We note that the uncertainty due to the size of the FRET probes was minimized by focusing on the change of the distances between the closed and open state, rather than absolute distances as discussed in previous section.

We note several limitations in the modeling: as the spring constant was kept constant through the simulations, resulting in a large force at beginning of the simulation, we applied both secondary structure restraints (*Trabuco et al., 2009*) and symmetry restraints (*Chan et al., 2011*) to prevent structural distortion. The secondary structure restraints prevents some subtle changes in the structure, such as kinks observed previously in the upper part of TM1 in the open model of MscL (*Deplazes et al., 2012*). Therefore, we limit our discussion of the open model to pore size and helix tilting. The membrane tension, which causes membrane thinning, plays an important role in the MscL opening process (*Corry et al., 2010*; *Louhivuori et al., 2010*; *Deplazes et al., 2012*). However, the restraint MD simulation cannot address the question of how the channel is activated. For the simplicity of the modeling, membrane tension is not considered here. We did observe that the membrane near the MscL becomes thinner during the channel opening process to match with the flattening MscL (*Figure 7—figure supplement 2*), confirming that a thinning membrane, likely caused by tension, matches the open channel better.

The resulting open state structure of MscL is shown in *Figure 7B,D*, and compared with the crystal structure of MscL in the closed state (*Figure 7A,C*). The open structure satisfies all the distance constraints measured in our smFRET experiments. In contrast, previous models based on ensemble FRET

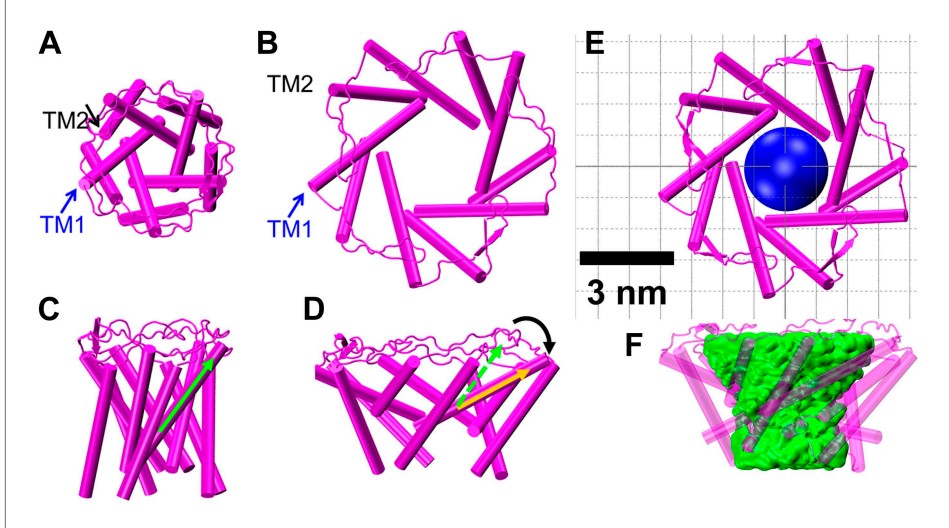

**Figure 7**. Model of the MscL structure in the open conformation. (**A** and **C**) The crystal structure of MscL in the closed state is shown for comparison (PDB: 2OAR [***Chang et al., 1998***; ***Steinbacher et al., 2007***]). (**B** and **D**) The structure of MscL in the open state (***Source Code 1 and 2***) was developed based on the smFRET measurements, satisfying all the distance constraints measured from smFRET experiments. In the open conformation, the pore is mainly lined by TM1 (indicated by blue arrows), consistent with the helix-tilt model. In addition, both TM1 and TM2 tilt toward the membrane plane (horizontal) upon channel activation, which is emphasized by the green and yellow arrows in the side views. The green arrows show the orientation of TM1 in the closed state while the yellow arrow indicated the orientation of TM1 in the open state. The angle between the two arrows is 27°. (**E**) A sphere with a diameter of 2.7 nm (blue) is shown in the MscL channel in the top view. (**F**) The surfaces of water molecules (green) inside the tunnel of MscL (magenta) are drawn and the narrowest constriction is ~2.7–2.8 nm.

The following figure supplements are available for figure 7:

**Figure supplement 1**. Developing a model for the open structure of MscL by inserting virtual springs.

**Figure supplement 2**. Side views of MscL open model (orange) in the POPC lipid bilayers.

measurements failed to be consistent with all experimental measurements (***Corry et al., 2010***). In the open conformation, the pore is mainly lined by helices TM1 (indicated by blue arrows), consistent with the helix-tilt model. In addition, it is observed that both TM1 and TM2 indeed tilt toward the membrane plane (horizontal) upon channel activation. For example, the orientation of TM1 tilts from the green arrow orientation (***Figure 7C***, closed state) to the yellow arrow orientation (***Figure 7D***, open state). The change in tiling angle of the TM1 and TM2 helices is $\Delta\theta_1 \approx 27°$ and $\Delta\theta_2 \approx 19°$, respectively, where θ is the angle between helix and the fivefold symmetry axis. The all-atom model and backbone model of the open state resulting from the current study are provided in PDB format in SI.

## Measurement of pore size in the open conformation

We used two independent methods to measure the pore size of MscL in the open state. The first method is to measure the movements of the residues forming the narrowest pore constriction of the channel, that is residues around I25 for *E. coli* MscL (***Chang et al., 1998***; ***Perozo et al., 2001***, ***2002a***; ***Corry et al., 2010***). However, this method, although straightforward, has its limitations. It is likely that the function of the channel is affected by mutation and labeling of (some of) the residues at the pore region. For example, the activation thresholds ($P_a$, defined as the pressure at which the first channel opening was observed [***Nomura et al., 2012***]) of mutants G22C and I24C are more than double the wild-type thresholds (***Figure 6***) and both ensemble and single molecule FRET measurements of these mutants showed no change in the FRET efficiency after adding 25% LPC. The effect of the point mutations near the pore on the electro-physiological properties of the channel can be quantitatively explained by the closed and open structure of MscL as shown in ***Figure 6—figure supplement 1***, the residue G22 (A20 in *M. tuberculosis* MscL) is very close to the pore and is facing the pore. The residue V22 (V22

in *M. tuberculosis* MscL) is also close to the pore and sandwiched between helix 1 and neighboring helix 1. Mutating these two residues is likely to perturb the channel function. On the other hand, the residue I25 is further from the pore than G22 and I24. The mutation I25C is less likely effect the channel properties. Indeed the I25C mutation does not affect the channel's gating parameters (*Figure 6C,D*). I25 is still close enough to the pore, making it a perfect candidate for measuring pore size. Furthermore, among the three mutated residues shown in *Figures 6*, I25 (green) is the only one facing outward from the channel axis and accessible from the periphery of the protein (*Figure 6—figure supplement 1B,D*). We were able to determine the movement of residue I25C (*Corry et al., 2010*); and measured that the residue I25 moves away from the pore center by $\Delta r = 1.2$ nm, indicating that the pore opens up by $\Delta D = 2.4$ nm in diameter. Taking into account that the pore diameter in the closed state ($\Phi_{close}$) is 0.4 nm (*Chang et al., 1998*), we conclude that the pore size in the open state ($\Phi_{open}$) is $\Phi_{open} = \Phi_{close} + \Delta D = 2.8$ nm, which agrees with previously reported values (*Perozo et al., 2002a*; *Corry et al., 2010*).

The second method is based on the open state model of MscL constructed by means of molecular dynamics. The surfaces of water molecules inside the channel were rendered (*Figure 7E,F*) using VMD (*Humphrey et al., 1996*) and the narrowest constriction seen provided an estimate of the pore size. This estimate accounts for all residues of the transmembrane domain and therefore is expected to be more accurate than the estimate of the first method. Using this method we estimate that the pore size of the MscL channel in the fully open state is 2.7–2.8 nm, which is consistent with the value from the first method, 2.8 nm.

## Discussion

We used a combination of experimental smFRET and computational modeling to study the conformational change of MscL upon channel activation. It is the first time that single molecule FRET has been applied to liposome-reconstituted membrane proteins with more than three monomers. We measured the distance changes of multiple residues from the MscL transmembrane α-helices (TM1 and TM2) during gating of the channel. For the first time, it is observed *directly* that both transmembrane helices swing away from the pore center, with rotational tilting involved. The results argue clearly in favor of the helix-tilt model. In addition, we developed by means of computational modeling a model of the channel structure in the open state based on the smFRET results and the crystal structure of the protein in the closed state as a reference. This model also confirms the helix-tilt model and yields a pore diameter of 2.8 nm. The smFRET experiments carried out in the present study observe MscL channels dynamics in lipid bilayers (liposomes) and not in detergents, which is a great advantage over crystallography that can result in different oligomeric states like those seen in the tetrameric structure of *S. aureus* MscL (*Liu et al., 2009*). It is possible that the detergent used in purification caused some portion (≤30%) of the MscL as hexamers, instead of the assumed pentamers. Nevertheless, our conclusion of the helix-tilt opening model is independent of the percentage of hexameric structure. However, the exact value of the open pore diameter would be slightly greater, 3.0 nm (30% hexamers), up to 3.3 nm (100% hexamers), still agreeing with previously reported values (*Sukharev et al., 2001b*; *Perozo et al., 2002a*; *Corry et al., 2010*).

The current study focused on the closed and fully open state of MscL. The fully open state was achieved by adding LPC to the liposomes (*Perozo et al., 2002a*, *2002b*). However, the technique introduced is not limited to these two states only. Single molecule FRET together with other techniques—for example, with patch-clamping done simultaneously—can answer many more questions than a crystal structure. For instance, it could probe the conformation of the channel during sub-conducting levels that involve partial MscL openings, or probe sequence of movements of the individual channel domains during opening of the channel.

## Materials and methods

### Mutation, expression, purification and labeling of MscL

The *E. coli* MscL gene (EcoMscL) was cloned into plasmid pQE-32 (Qiagen, Hilden, Germany) as the *Bam*HI-*Sal*I fragment, which also added a hexa-histidine tag (his-tag) to the protein at the N-terminus. The protein was expressed in *E. coli* (M15 strain) (Qiagen) that were lysed by sonication and purified from DDM solubilized membranes using TALON Metal affinity chromatography (Clontech Laboratories, Inc, Mountain View, CA), followed by a further purification step using fast protein liquid chromatography (FPLC; Superdex 200 10/300 GL column, GE Healthcare, Pittsburgh, PA). Purification was performed in the presence of 1 mM DDM.

The wild type of MscL protein does not contain any cysteine. To label the proteins with fluorescent probes, MscL was mutated using site-directed mutagenesis such that a residue at the desired position was replaced by a cysteine. Because the MscL protein is a homo-pentamer (*Chang et al., 1998*), this mutation introduced five identical cysteine sites.

The protein with his-tag was then labeled with Alexa Fluor 488 (AF488) and/or Alexa Fluor 568 (AF568) maleimide, which specifically reacted with the introduced cysteines (*Kim et al., 2008*). Right before labeling, proteins were reduced with 10 mM DTT for 30 min, followed by purification using PD-10 desalting columns (GE Healthcare). We titrated the pentameric protein-to-fluorophore molar ratio from 1:1 to 1:5 and used the molar ratio of 1:5 for labeling in all the experiments. Under our labeling conditions, this ratio gave satisfying results such that most of the proteins are labeled (averagely ~1.7 donors and ~1.3 acceptors per pentamer) and that many of proteins are attached by a single donor and a single acceptor (~30% of good traces show multiple donors and/or acceptors). Excess fluorophores were then removed using PD-10 desalting columns. The sample was reduced with 10 mM DTT before this purification step. A note to make is that the fluorophores (Alexa Fluor 488 male-imide and Alexa Fluor 568 maleimide) come as mixtures of 5′ and 6′ isomers, which would potentially complicate interpretation of smFRET data. However, we expect that the results would not be affected because the exactly same fluorophores have been successfully used in many single molecule FRET studies (*Marras et al., 2002*; *Yin et al., 2005*; *Jäger et al., 2006*; *Granier et al., 2007*; *Majumdar et al., 2007*).

## Reconstitution and opening of MscL in liposomes

MscL channels were reconstituted into artificial liposomes (~50 nm diameter), following the protocol described in *Perozo et al. (2002a, 2002b)*. Liposomes were prepared by drying, rehydrating and extruding lipids through filters with ~50 nm pores. The lipids used in all the measurements were a mixture of 1-palmitoyl-2-oleoyl-sn-glycero-3-phosphocholine (POPC, Avanti Polar Lipids, Inc, Alabaster, AL) and 1,2-dioleoyl-sn-glycero-3-phosphoethanolamine-N-biotinyl (BPE, Avanti Polar Lipids, Inc.) dissolved in chloroform at a molar ratio of POPC:BPE = 1000:20. BPE was used for immobilization. To incorporate MscL channels into the liposomes, a mixture of unlabeled and labeled MscL proteins (5% labeled) was then reconstituted into the liposomes, at a final volume of 1 ml, with a protein/lipid (molar) ratio of 1:200, resulting in a molar ratio of 1:4000 for the labeled proteins to lipids. The liposomes were immobilized onto a glass coverslip. This immobilization was achieved by biotin-avidin linkages between biotinylated-PEG molecules on the surface to a neutravidin molecule, and then biotinylated lipids (BPE) in the liposomes (*Roy et al., 2008*).

To open the MscL channels in the liposomes, a conical lipid, 1-oleoyl-2-hydroxy-sn-glycero-3-phosphocholine or lysophosphatidylcholine (LPC, Avanti Polar Lipids, Inc), was added to the liposomes, at a molar fraction of 25%. As LPC incorporates itself into the outer leaflet of a lipid bilayer, it introduces membrane tension, changes the lipid pressure profile, and triggers the MscL to open (*Perozo et al., 2002a*, *2002b*).

## Electrophysiological recording

MscL protein purification and reconstitution into soybean azolectin liposomes were described previously (*Nomura et al., 2012*). All results were obtained with proteoliposomes at the protein: lipid ratio of 1:200 (wt/wt). Channel activities of the wild-type and mutant MscL were examined in inside-out liposome patches using patch-clamp technique. Borosilicate glass pipettes (Drammond Scientific Co, Broomall, PA) were pulled using a Narishige micropipette puller (PP-83; Narishige, Tokyo, Japan). Pipettes with resistance of 2.5–4.9 MΩ were used for the patch-clamp experiments. Pipette and bath solution contained 200 mM KCl, 40 mM MgCl2, and 5 mM HEPES (pH 7.2 adjusted with KOH). The current was amplified with an Axopatch 200B amplifier (Molecular Devices, Sunnyvale, CA), filtered at 2 kHz and data acquired at 5 kHz with a Digidata 1440A interface using pCLAMP 10 acquisition software (Molecular Devices, Sunnyvale, CA) and stored for analysis. Negative pressure (suction) was applied to the patch pipettes using a syringe and was monitored with a pressure gauge (PM 015R, World Precision Instruments, Sarasota, FL).

## Selection of MscL with a single donor and a single acceptor

Since the MscL channel is a homo-pentamer (*Chang et al., 1998*) (or possibly homo-hexamer [*Gandhi et al., 2011*]), there is always a distribution of various donor/acceptor combinations. To exclude signal from those channels having multiple donors or multiple acceptors, the fluorescence intensity of single channels (and hence the step-wise photobleaching) was monitored. Because multiple donors or acceptors

have multiple 'staircase' photobleaching, these channels were simply not used. Only the traces with a clear single-step photobleaching in both donor and acceptor channels were included in the analysis. Subtraction of the intensities (averaged) before and after photobleaching gives the intensities of donor ($I_D$) and acceptor ($I_A$), which are then used for FRET efficiency calculation as described below.

## Single molecule FRET measurement

Single molecule FRET experiments were performed using total internal reflection fluorescence microscopy (TIRFM) with a 1.45 NA 100X oil immersion objective (**Selvin and Taekjip, 2007**; **Roy et al., 2008**). The fluorescence intensities were used to calculate the energy transfer efficiency by the corrected FRET equation: $E = (I_A − lI_D)/(I_A + \gamma I_D)$: where E is the FRET efficiency, l represents leakage of donor signals in the acceptor channel, $\gamma$ is the correction factor which accounts for the differences in quantum yield and detection efficiency between the donor and the acceptor, $I_A$ and $I_D$ represent the acceptor and donor intensities, respectively (**Roy et al., 2008**). Note that the direct excitation of the acceptor by the donor excitation has been corrected automatically when getting the acceptor intensity from the fluorescence traces. The distance between the donor and acceptor is given by $R = R_0(E^{−1}−1)^{1/6}$, where $R_0$ is

the Förster radius (**Förster, 1948**). The Förster radius, $R_0$, given by $R_0 = \left( \dfrac{0.529 \kappa^2 \, Q_D \, J(\lambda)}{N_A \, n^4} \right)^{1/6} \propto \left( \kappa^2 Q_D \right)^{1/6}$,

and its error were measured experimentally by measuring the absorbance and fluorescence spectra, quantum yield of the donor, AF488, ($Q_D = Q_{AF488}$) and anisotropy ($A_a$ and $A_d$ which give the maximum possible error in $\kappa^2$) of the fluorescent probes conjugated to proteins.

## Measurement of quantum yield of AF488 conjugated to MscL

The quantum yield of AF488 conjugated to MscL was measured using fluorescein in 0.1 M NaOH as a standard (**Fery-Forgues and Lavabre, 1999**; **Lakowicz, 1999**) using the equation $Q_X = \dfrac{A_S}{A_X} \times \dfrac{F_X}{F_S} \times \left( \dfrac{n_X}{n_S} \right)^2 \times Q_S$, where Q is the quantum yield, A is the absorbance at the excitation wavelength (470 nm); F is the area under the corrected emission curve, and n is the refractive index of the solvent. Subscripts S and X refer to the standard (fluorescein) and to the unknown (AF488), respectively. The spectra of absorbance and fluorescence of AF488-MscL in PBS + DDM (1 mM DDM) were measured using Agilent 8453 UV-Vis absorbance spectrophotometer (Agilent Technologies, Santa Clara, CA) and PC1 spectrofluorimeter (ISS, Inc., Champaign, IL), respectively.

## Measurement of anisotropy of fluorophores conjugated to MscL

In order to determine the maximum error in the orientation factor, $\kappa^2$, and therefore the error in $R_0$, the anisotropy of the fluorophores conjugated to MscL was measured. The fluorophores-protein conjugates were immobilized on a glass coverslip which was covered with PEG (5% biotinylated), then a layer of neutravidin (Thermo Scientific, Waltham, MA), followed by a layer of penta-his biotin conjugate (Qiagen). The emission of the fluorophores-protein conjugates were split into two channels of polarization and used to calculate the anisotropy, $A = \dfrac{I_{\parallel} − I_{\perp}}{I_{\parallel} + 2I_{\perp}}$, where $I_{\parallel}$ is the fluorescence emission with polarization parallel to the excitation polarization and $I_{\perp}$ is the fluorescence emission with polarization perpendicular to the excitation polarization (**Lakowicz, 1999**). Anisotropies were corrected for the intrinsic polarization properties of the microscope by calibrating to known freely diffusing fluorophores. Anisotropies were also corrected for the high numerical aperture of the objective. Then the maximum range of $\kappa^2$ was given by $\kappa^2_{max} = 2/3(1 + 2.5A_d+2.5A_a)$ and $\kappa^2_{min} = 2/3(1−1.25A_d−1.25A_a)$ where $A_d$ and $A_a$ are the anisotropy of AF488 (donor) and AF568 (acceptor), respectively (**Dale et al., 1979**; **Cha et al., 1999**).

## Estimatating the sizes of fluorescent probes

To evaluate directly the sizes of the fluorescent probes used in our FRET experiments, the molecular structures of the AF488-C5-Maleimide and AF568-C5-Maleimide were constructed using Avogadro (**Hanwell et al., 2012**). Both the 5'- and 6'-isomers were constructed. These structures were then optimized in Avogadro with molecular dynamics using the universal force field (UFF) (**Rappe et al., 1992**). From the optimized molecular structures (shown in **Figure 4—figure supplement 1**), we estimated the probe sizes which were defined as the distance between the oxygen atom of the fluorophore (indicated by the magenta arrows in **Figure 4—figure supplement 1B,F**) and the nitrogen atom of the maleimide group (indicated by the cyan arrows in **Figure 4—figure supplement 1A–E**). We found that the donor is

17.1 Å (5′-isomer) or 16.3 Å (6′-isomer) while the acceptor is 17.4 Å (5′-isomer) or 17.4 Å (6′-isomer). The difference in the molecular size between donor-isomers or between acceptor-isomers is small, <5%.

## Modeling the MscL open structure through restraint molecular dynamics (MD) simulation

Due to lack of an *E. coli* MscL (EcoMscL) crystal structure, the simulation were performed using the structure of MscL from *M. tuberculosis* (MtMscL, PDB: 2OAR) (*Chang et al., 1998*; *Steinbacher et al., 2007*). The CP domain was truncated in the simulation because the complete deletion of the CP does not change the gating parameters substantially (*Anishkin et al., 2003*). The residues to which the distance constraints were applied, were shifted according to the sequence alignment in *Chang et al. (1998)*. A spring constant of 0.2 kcal mol$^{-1}$Å$^{-2}$ was used for the virtual spring in the distance constrained simulation. Both secondary structure restraints (*Trabuco et al., 2009*) and symmetry restraints (*Chan et al., 2011*) were applied to prevent structural distortion under large force in the distance constrained simulation. Total simulation time is 5 ns. A model of MscL in the open state was obtained at the end of the distance constrained simulation, when the simulation satisfied all the distance constraints measured by means of smFRET experiment. The restraint MD simulation procedure is similar to the one used previously (*Corry et al., 2010*; *Deplazes et al., 2012*).

The simulation system was prepared by first imbedding the crystal structure of MscL (PDB: 2OAR) (*Chang et al., 1998*; *Steinbacher et al., 2007*) into a membrane patch with 1727 POPC lipids. Solvent was then added to both sides of the membrane, and the system was neutralized with 200 mM NaCl using VMD (*Humphrey et al., 1996*). The final simulation system contained 1,137,413 atoms. The all-atom MD simulations were performed using NAMD 2.9 (*Phillips et al., 2005*) with the TIP3P model (*Jorgensen et al., 1983*) for explicit water and the CHARMM36 force field (*Best et al., 2012*). The simulation was conducted in the NPT ensemble (constant pressure and temperature) with periodic boundary condition. Constant temperature of 300 K was maintained using a Langevin thermostat with a damping coefficient of 1 ps$^{-1}$. A Nosé–Hoover Langevin piston barostat was used to maintain a constant pressure of 1 atm with a period of 200.0 fs and damping timescale of 100.0 fs. The multiple time-stepping algorithm was employed, with an integration time step of 2 fs, the short-range force being evaluated every time step, and the long-range electrostatics every second time step. Non-bonded energies were calculated using particle mesh Ewald full electrostatics and a smooth (10–12 Å) cutoff of the van der Waals energy.

## Acknowledgements

The authors acknowledge supercomputer time on Stampede provided by the Texas Advanced Computing Center (TACC) at The University of Texas at Austin through Extreme Science and Engineering Discovery Environment (XSEDE) Grant MCA93S028, supported by National Science Foundation Grant OCI-1053575. We thank Kai Wen Tseng for assistance with the quantum yield measurements. We also thank Eduardo Perozo (U of Chicago) for early work and for some MscL plasmids.

## Additional information

### Funding

| Funder | Grant reference number | Author |
|---|---|---|
| National Institutes of Health | R01 GM068625 | Paul R Selvin |
| National Institutes of Health | R01 GM067887, U54 GM087519, 9P41GM104601 | Klaus Schulten |
| National Science Foundation | PHY0822613 | Paul R Selvin, Klaus Schulten |
| National Health and Medical Research Council | 635525 | Boris Martinac |

The funders had no role in study design, data collection and interpretation, or the decision to submit the work for publication.

## Author contributions

YW, Expressed and purified proteins, Performed single molecule FRET measurements, Conception and design, Analysis and interpretation of data, Drafting or revising the article; YL, Performed molecular dynamics simulations, Analysis and interpretation of data, Drafting or revising the article; HAD, Expressed and purified proteins, Analysis and interpretation of data; TN, Performed electrophysiological experiments, Conception and design, Analysis and interpretation of data; MTH, Initiated the project, Conception and design; PRR, Acquisition of data; KS, Performed molecular dynamics simulations, Conception and design; BM, Performed electrophysiological experiments, Conception and design; PRS, Conception and design, Drafting or revising the article

## Additional files

### Supplementary files

• Source code 1. All-atom model of the open structure of MscL developed in this study.
• Source code 2. Backbone model of the open structure of MscL developed in this study.

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
