## [Decision Letter]

Thank you for sending your work entitled “Single Molecule FRET Reveals Pore Size and Opening Mechanism of MscL” for consideration at *eLife*. Your article has been favorably evaluated by a Senior editor, John Kuriyan, and 2 reviewers, one of whom is a member of our Board of Reviewing Editors.

The Reviewing editor and the other reviewer discussed their comments before we reached this decision, and the Reviewing editor has assembled the following comments to help you prepare a revised submission.

In this paper the authors use single-molecule FRET measurements (smFRET) to analyze the degree and nature of the opening of a mechanosensitive channel in a lipid bilayer environment. The smFRET measurements permitted the authors to estimate the change in the distance between donors and acceptors from the closed to open states. To quantitatively investigate in detail how the MscL channel opens, the authors developed a computational model for the open state, starting from the crystal structure of MscL in the closed state (PDB: 2OAR) employing the measured residue movements. For this purpose, MD simulations with distance constraints were performed. This is an impressive piece of work because it analyzes the channel in a lipid bilayer environment rather than in detergent micelles, and because the authors use photobleaching to ensure that they are studying channels with one donor and one acceptor, thus avoiding complications that might otherwise undermine the analysis. The importance of the work arises because there is no crystal structure for an open form of these channels, and so the paper potentially fills in major gaps in understanding.

Major concerns:

1) It is emphasized that the modeled open structure resulting from the molecular dynamics satisfies all the distance constraints derived from their smFRET experiments. But this statement leaves out completely any uncertainty due to the size of the FRET probes themselves. It is stated that, for each measured residue, ten virtual springs were placed, five springs between identical residues from adjacent monomers and five springs between identical residues from non-adjacent monomers. The equilibrium lengths of the springs were chosen by adding the distance changes measured from smFRET to the equilibrium distances seen in the closed state. However, it is not specified between which atoms the virtual spring are introduced.

2) The discussion of the maximum possible errors in R0 is extensive, but mainly concerns the experimental FRET aspects of the problem. However, there are additional difficulties in trying to convert the measurements into a structural model. In fact, converting the changes in distance into a structural model is not straightforward. The channels were labeled with Alexa Fluor 488 (AF488) and Alexa Fluor 568 (AF568). The smFRET distances report the donor-acceptor distances. To model this correctly, it is necessary to account for the size and length of the probes.

3) The authors should discuss, on the structural level, the expected effect of inserting a molecule of the size of AF488 or AF568.

4) The binding of one donor and one acceptor molecule per pentamer will break the five-fold symmetry. In what way does this affect the geometric construction in Figure 5?

5) Can the activity of the mutants from Figure 6 be explained/rationalized with the final model?

---

## [Author Response]

*1) It is emphasized that the modeled open structure resulting from the molecular dynamics satisfies all the distance constraints derived from their smFRET experiments. But this statement leaves out completely any uncertainty due to the size of the FRET probes themselves. It is stated that, for each measured residue, ten virtual springs were placed, five springs between identical residues from adjacent monomers and five springs between identical residues from non-adjacent monomers. The equilibrium lengths of the springs were chosen by adding the distance changes measured from smFRET to the equilibrium distances seen in the closed state. However, it is not specified between which atoms the virtual spring are introduced*.

In the computational models, the virtual springs were introduced between center carbon atoms (Cα) of residues in measured smFRET. This information is provided in the revised manuscript. The virtual springs were not applied to side chains because the flexibility of side chains will likely introduce errors under large forces in the modeling process. The uncertainty due to the size of the FRET probes was minimized by focusing on the change of the distances between the closed and open state, rather than absolute distances. More details are discussed in response to reviewers’ major concerns 2 and 3. The discussions are also added to the revised manuscript.

*2) The discussion of the maximum possible errors in R0 is extensive, but mainly concerns the experimental FRET aspects of the problem. However, there are additional difficulties in trying to convert the measurements into a structural model. In fact, converting the changes in distance into a structural model is not straightforward. The channels were labeled with Alexa Fluor 488 (AF488) and Alexa Fluor 568 (AF568). The smFRET distances report the donor-acceptor distances. To model this correctly, it is necessary to account for the size and length of the probes*.

We thank the reviewers and agree with the reviewers that the size and length of the probes brings additional difficulty to converting the FRET measurements to *absolute* distances and to structural modeling. To overcome the difficulty, our strategy, as we emphasized in the manuscript, is to focus on the movement of residues, instead of the absolute distances, and then to develop the structural model based on the changes of distances.

For illustration purpose, we set up the general scene in Figure 8 first. The MscL pentamer protein (light blue), centered at O, is labeled with two probes (dark blue) on two cysteine residues (specified by the residue #), while the other three residues (with the same residue # but on different subunits) remain empty / unlabeled (dashed green). (Here the breaking of symmetry, mentioned by the reviewers in major concern 4, has been taken into account in this general case.) To emphasize the size and length of the probes, the actual chromophores are indicated by green and red filled circles. Then the actual chromophores of the labeled probes (green and red filled circles) and the center of the protein (O) define a pentagon (dotted orange pentagon, with a side length of R) and a circumcircle (black circle, with a diameter of D, or a radius of r = D/2). We call R, D, and r, apparent distance/diameter/radius because they could be measured/calculated from FRET experiments (with appropriate corrections) and they gave the apparent size of the protein (e.g., the protein appears to have a radius of r based on FRET experiments).Author response image 1.Geometric analysis of the distances of interest.

There is another set of distances, which are more relevant to the protein and to the MD simulations for the structural model. We call this set of distances true values. For example, the true radius of the protein (r_t_) (referring to a specific residue; corresponding to the red circle) is defined as the distance between the Cα atom of the residue and the center of protein (O).

Due to the finite size of the probes (r_p_ ≠ 0, Figure 8), generally we have r = r_t_ + r_p_ ≠ r_t_. As a result, converting the FRET measurements into a structural model is not straightforward. However, if the size of the probes does not change (i.e., Δ*r*_*p*_ = 0) upon channel activation, we then have Δ*r* = Δ*r*_*t*_ + Δ*r*_*p*_ = Δ*r*_*t*_. We believe Δ*r*_*p*_ = 0 is a reasonable assumption for the following reasons: (1) no chemical reactions are happening for the probes and thus the structures of the probes do not change before and after channel opening; (2) anisotropy measurements show that the orientation of probes are not constrained significantly. As a result, the change in the apparent distances is the same as the change in the true distances. In other words, the movement of residues in the radical direction (Δ*r*_*t*_) can be obtained from the FRET measurements, even if the sizes and lengths of probes are nonzero.

A note to make is that we have assumed that donors and acceptors have similar sizes in the argument above. We argue that this assumption is valid. To justify it, molecular structures of th e probes used in our experiments (AF488-C5-Maleimide and AF568-C5-Maleimide) were constructed using Avogadro software (28). The structures were then optimized in Avogadro with molecular dynamics using the universal force field (UFF) (49) (Figure 9; we show only the 5’ isomers here; please see Figure 10 for 6’ isomers). From the optimized molecular structures, we estimated the probe sizes, which were defined as the distance between the oxygen atom of the fluorophore (indicated by the magenta arrows in Figure 9) and the nitrogen atom of the maleimide group (indicated by the cyan arrows in Figure 9). We found that the donor is 17.1 Å (Figure 9) while the acceptor is 17.4 Å (Figure 9). The difference between the probe sizes is < 2%, confirming that the sizes of donors and acceptors are very similar.Author response image 2.Molecular structures of fluorescent probes used in the smFRET experiments. (A, B) Alexa Fluor 488; (C, D) Alexa Fluor 568.Author response image 3.Molecular structures of fluorophores used in the experiments. (A, B) 5’ isomer of Alexa Fluor 488; (C, D) 6’ isomer of Alexa Fluor 488; (E, F) 5’ isomer of Alexa Fluor 568; (G, H) 6’ isomer of Alexa Fluor 568.

On the other hand, we also note that there are possible situations where the size of the probe can change (i.e., Δ*r*_*p*_ ≠ 0) upon channel opening. For example, the probe might be sterically hindered and the orientation is constrained. Although the chance is not high (please see the reply to major concern 3), this possibility cannot be ruled out. Therefore, the authors now explicitly state the assumption in the manuscript.

In the resubmission, we emphasize once again that what really matters is the movement of residues. We have modified the manuscript to illustrate it in much more detail than the previous version.

*3) The authors should discuss, on the structural level, the expected effect of inserting a molecule of the size of AF488 or AF568*.

We thank the reviewers for the suggestion. We include a brief discussion on the expected effect of inserting of fluorophores in the resubmission as follows.

Because the size of both Alexa fluorophores is significant (∼1.7 nm), it is possible that the attachment of the fluorophores to MscL channel results in various effects on the protein and on the FRET measurements. For example, the presence of the fluorophores might sterically hinder the conformational change of the proteins and prevent them from opening or closing. Also, the steric hindrance might constrain the orientation of fluorophores, affect the relative orientation between the fluorophores and therefore add more errors on the distances converted from FRET efficiencies. In addition, the insertion of fluorophores to the protein might force the channel to be in a state different from the fully closed state, resulting in the measured distance change being underestimated.

However, we would like to emphasize that the expected effect is not significant for the following reasons. First, if the insertion of fluorophore would result in significant steric hindrance on the protein, it is expected that the labeling is difficult (i.e., it takes much more effort for the fluorophores to be attached due to the steric hindrance). In other words, it is expected that steric hindrance is not significant on the mutants that are labeled well. More importantly, the channels after being labeled with AF488 and AF568 were confirmed to be functional by both ensemble FRET experiments and patch-clamp measurements (for example, see Figure 6 and previous publications of the co-authors (17)).

*4) The binding of one donor and one acceptor molecule per pentamer will break the five-fold symmetry. In what way does this affect the geometric construction in*
Figure 5?

The breaking of the five-fold symmetry due to the binding of one donor and one acceptor per pentamer will not affect the geometric construction in Figure 5. However, some of the (absolute) distances need re-defining. Nonetheless, as shown in the reply to major concern 2, the changes of distances will not be affected and the developed structural model remains valid.

We now put a brief comment to address this concern in the resubmission.

*5) Can the activity of the mutants from*
Figure 6
*be explained/rationalized with the final model*?

We thank the reviewers for this great question. We have checked where the mutated residues (A20, V22, and I23 for MD simulations, which are corresponding to G22, I24, and I25, respectively, in *E. coli* MscL for smFRET experiments) in Figure 6 located in the *Mycobacterium tuberculosis* MscL structure. The effect of the mutations can be explained qualitatively based on the closed state as seen in the crystal structure and the open state of our final model. As shown in Figure 11, the residue A20 (red) is the closest to the pore among the three residues and facing the pore. The residue V22 (blue) is the second closest to the pore and sandwiched between helix 1 and neighboring helix 1. Mutating these two residues (A20 and V22) is likely to perturb the channel function and electro-physiological properties. On the other hand, the residue I23 (green) is farthest from the pore among the three residues. The mutation I23C is less likely effect the channel properties compared to mutating A20 or V22. I23 is still relatively close to the pore compared to the rest of the residues in MscL, making it a perfect candidate for measuring pore size. Furthermore, among the three mutated residues shown in Figure 6, I23 (green) is the only one facing outward from the channel axis and accessible from the periphery of the protein (see Figure 11). We have added this explanation in the revised manuscript and Figure 11 is provided in as Supplementary Figure 10.Author response image 4.Positions of MtMscL mutants at the proximity of the narrowest pore constriction. (A) Top view and (B) side view of the close state as seen in the crystal structure. (C) Top view and (D) side view of the open state modeled from smFRET measurement. Three residue are shown: A20 (red), V22 (blue), and I23 (green), which are equivalent to G22, I24, and I25, respectively, in *E coli* MscL. A20 and V22 are closer to the pore than I23. I23 is the only one among then that is facing outward from the channel axis and is accessible from the periphery of the protein.